# Detection and identification of vacancy defects in antimony selenide

David J. Keeble [1] ✉, Theodore D. C. Hobson [2], Julia Wiktor [3], Ethan Berger [3], Marcel Dickmann [4], Mohamed R. M. Elsharkawy [1,5], Werner Egger[4], Jonathan D. Major [2] & Ken Durose [2]

Antimony selenide ($Sb_2Se_3$) has an optimal bandgap and absorption coefficient for thin film solar cell applications and comprises earth abundant elements. The rate of increase in reported power conversion efficiencies has slowed due to a persistently large open circuit voltage deficit attributed to detrimental concentrations of point defects. Here we use depth-profiling positron annihilation lifetime spectroscopy to study $Sb_2Se_3$ crystals and thin films. The method is specific to neutral and negative charge states of vacancy-related defects. Both monovacancy and divacancy defects are identified in intrinsic and n-type samples but no monovacancy defects are detected in the p-type sample. Comparison of the experimental positron lifetimes with density functional theory calculated values provide evidence for the observation of Sb monovacancies in the −3 state and of Se monovacancies in the −2 state. The results are consistent with recent density function theory predictions that the Sb and the Se monovacancy defects both have accessible negative charge states.

The global demand for sustainable, affordable, photovoltaic (PV) materials that can enable thin film solar cells with power conversion efficiency (PCE) greater than 20 % motivates the search for suitable earth abundant PV materials[1]. The binary semiconductor $Sb_2Se_3$ has attracted particular attention due to its appropriate bandgap, 1.18 eV, high optical absorption coefficient and carrier mobility and the low toxicity and high natural abundance of the constituent elements. Antimony selenide solar cell PCEs have increased from, for example, 2.3 % in 2014 to 7.6% in 2018, then from 9.2% in 2019 to 10.1 % in 2022[2–5]. But these values remain far from the detailed-balance limit of ~30%[6]. Performance of current $Sb_2Se_3$ cells is inhibited by the low values of open-circuit voltages, $V_{oc}$, which are typically only ~40 % of the bandgap value (i.e. ~50% of the Shockley-Queisser limit). This $V_{oc}$ deficit results from the presence of point and extended defects and associated carrier recombination[7,8]. Deep-level transient spectroscopy (DLTS) and temperature dependent admittance spectroscopy (TAS)

measurements clearly show the presence of defects in monocrystalline and thin film $Sb_2Se_3$[4,9–13]. These methods can identify charge transition energy levels within the bandgap due to defects, but they do not provide information on the chemical nature and local structure. This can only be inferred by comparison with density functional theory (DFT) calculations of charge transition levels for specific point defect types. Currently, there is no experimental identification of intrinsic point defects in $Sb_2Se_3$ using local structure sensitive spectroscopy methods. Here we report the detection of monovacancy and divacancy defects in $Sb_2Se_3$ crystals and thin films using the vacancy-related defect specific characterization technique positron annihilation lifetime spectroscopy performed using a depth-profiling high intensity positron beam.

Antimony selenide consists of strongly bonded quasi-one-dimensional (quasi-1D) $[Sb_4Se_6]_n$ ribbon units stacked together by weak van der Waals interactions forming an orthorhombic crystal

[1]Physics, SUPA, School of Science and Engineering, University of Dundee, Dundee, UK. [2]Stephenson Institute for Renewable Energy, Department of Physics, University of Liverpool, Liverpool, UK. [3]Department of Physics, Chalmers University of Technology, Gothenburg, Sweden. [4]Institut für Angewandte Physik und Messtechnik, Universität der Bundeswehr München, Neubiberg, Germany. [5]Physics Department, Faculty of Science, Minia University, Minia, Egypt. ✉e-mail: d.j.keeble@dundee.ac.uk

structure (Fig. 1)[14]. This quasi-1D structure has the potential to enable structural reconstructions that result in a suppression of deep electronic states associated with grain boundaries, in stark contrast to other covalently-bonded semiconducting photovoltaics[15,16]. The low lattice symmetry results in two inequivalent Sb sites and three inequivalent Se sites within the ribbon units (Fig. 1). In consequence, the properties of the six intrinsic point defects, two monovacancies, $V_{Se}$ and $V_{Sb}$, two antisites, $Sb_{Se}$ and $Se_{Sb}$, and two interstitials, $Se_i$ and $Sb_i$, can vary with site, resulting in complex defect physics and chemistry.

First principles calculations provide essential insight on possible specific point defects, enabling determination of charge transition energy level positions, defect formation energies, and detailed local structure[6,17–22]. The defect formation energies then allow theoretical values for defect concentrations under given growth conditions to be calculated. This information, in turn, can enable detailed modeling of photovoltaic devices. However, it is desirable to validate the predictions of DFT against the results from point defect local structure sensitive experimental methods, for example, positron annihilation spectroscopy and electron magnetic resonance spectroscopy.

The defect formation energies and possible charge transition energy levels are determined by the nature of the bonding and local structural rearrangements of neighbour atoms adopted in the ground state configurations of the defects. Recently, it has been clearly shown that the assumed point defect ground state configurations commonly obtained from first-principles calculations can, in fact, be metastable arrangements[22]. The conventional approach starts from a high symmetry initial geometry and uses a gradient-based optimization algorithm which increments the neighbour atom positions seeking a minimum in the potential energy surface, but this may be a local minimum. A more sophisticated search procedure that involves increasing the number of nearest neighbour atoms that are displaced and that varies the amount of distortion, and that applies random perturbations to the coordinates of all the atoms in the supercell, has been developed[22]. In particular, it is found necessary to apply this approach for defects in materials with low-symmetry structures and flexible bonding environments, as exemplified by $Sb_2Se_3$ and $Sb_2S_3$[6,22]. These studies cause significant changes in the predicted nature of the possible defect charge transition levels and their positions within the bandgap when compared to previous reports, and hence change the interpretation of the experimentally determined defect levels.

The removal of an Sb atom with its three donating electrons creates a neutral vacancy, $V_{Sb}^0$, but leaves three holes on the neighbour Se dangling bonds. Localization of three electrons at the vacancy compensate these and then results in a stable, fully ionized, $V_{Sb}^{-3}$ charge state. Removing a Se ion results in a fully ionized $V_{Se}^{+2}$ state[6]. Importantly, however, other charge states can be stabilized by the localization of excess charge at the vacancy site or through local atomic rearrangements, or a combination of both. Density function theory can provide detailed information on the nature of these processes and hence on the type and position of possible charge transition energy levels within the energy bandgap. The recent calculations applying the SHAKENBREAK approach to the monovacancy defects in $Sb_2Se_3$ find that both the Sb and Se vacancies are amphoteric[6]. It is reported that both monovacancy defects are stable in positive charge states for Fermi-level positions below mid-bandgap and that both have stable negatively charged states for positions close to the conduction band[6]. They also predict structural reconstructions involving either Se or Sb trimers that enable the stabilization of an abnormal positive +1 state for $V_{Sb}$ or a –2 state for $V_{Se}$, respectively[6]. Further, the Sb vacancies at the two possible sites (Fig. 1) are reported to both exhibit an unusually four-electron negative-$U$ transition giving a +1/–3 level near midgap. The Se vacancy at site 2 exhibits a similar four-electron transfer, giving a +2/–2 transition level in the upper half of the bandgap. The Se vacancies at the other two sites exhibit two sequential two-electron negative-$U$ transitions giving a +2/0 and 0/–2 pair of levels. In consequence, it is predicted that site-1 and site-3 Se monovacancies have an accessible neutral charge state for a narrow range of Fermi-level positions just above midgap[6]. By contrast, earlier DFT calculations predicted a more conventional ordering of charge transition levels within the bandgap where the Sb vacancies are stable in the –3 state for Fermi-level positions through almost the whole bandgap, with no accessible positive states, and where Se vacancies exist in positive charge states for Fermi-level positions lower in the bandgap but in the neutral charge state for level positions close to and above midgap and have no accessible negative charge states[18–21]. These differences in position and nature of the vacancy defect charge transition levels directly affect previous studies that rely on comparison with first-principles calculations to enable experimentally determined levels to be interpreted in terms of the presence or absence of specific vacancy defects[9,23,24].

Vacancy defects that are stable in neutral or negative charge states can be detected and identified by positron annihilation spectroscopy[25–27]. These methods have unique specificity for vacancy, open volume, defects and have a sensitivity limit of approximately $10^{15}$ cm$^{-3}$ and can estimate vacancy defect concentrations up to approximately $10^{19}$ cm$^{-3}$ beyond which saturation positron trapping occurs[25,26]. Positron annihilation lifetime spectroscopy (PALS) enables the experimental detection of multiple positron states. Positrons implanted into perfect material enter a Bloch state and annihilate with a lifetime characteristic of the overlap between the positronic and electronic densities, termed the perfect lattice or bulk lifetime, $\tau_B$. Missing atom sites present a trapping potential for positrons, but the local charge of the vacancy centre with respect to the lattice is also of critical importance in determining positron trapping. The Coulomb barrier presented by positively charged vacancy defects results in negligible trapping rates, while, in contrast, neutral or negatively charged vacancy defects are strong positron traps. Positron states localized at vacancy defects have lifetimes that are longer than that for the perfect lattice state due to the lower electron density at the missing atom site[25–27]. The lifetime of a positron state localized at specific vacancy defect is a characteristic of the particular defect type. Observation of both the perfect lattice state annihilations and longer lifetime defect localized state annihilation events enables the trapping to vacancy-related defects to be unambiguously established. Comparison of the experimentally determined lifetimes to those predicted by two-component DFT (TC-DFT) calculations for the positron states localized at specific vacancy–related defects can allow identification[26,28]. Moreover, positron lifetime measurements performed using a high intensity variable energy (VE) positron beam enable depth–profiling of vacancy–related defects from the near-surface down to depths of a few microns[29,30].

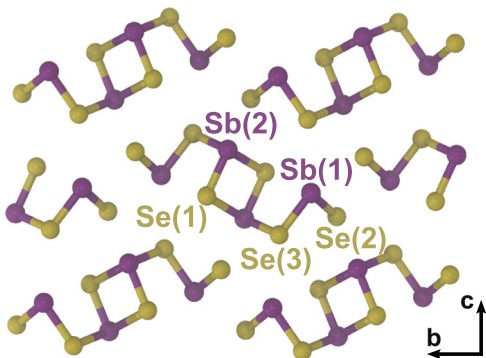

**Fig. 1 | Crystal structure of Sb₂Se₃.** Showing the ribbon elements extending along the a-axis (space group 62, *Pnma* setting) and including atom site labels.

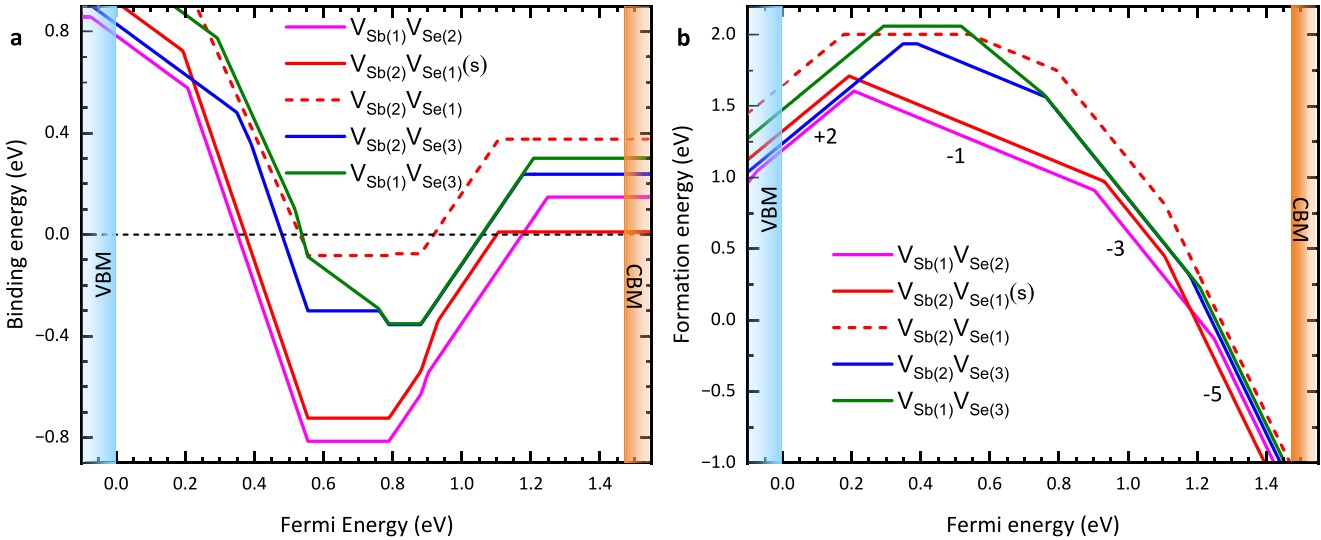

**Fig. 2 | Binding and formation energies of divacancies. a** Binding energy of the 5 possible divacancies. The $(V_{Sb(1)}V_{Se(2)})^{-1}$ and the short-bond (s) configuration of $(V_{Sb(2)}V_{Se(1)})^{-1}$ are found to have the lowest binding energies with values of −0.81 eV and −0.72 eV, respectively. **b** The associated formation energy. Labels show the charge states of the divacancy with the lowest formation energy. The positions of the valence band maximum (VBM) and conduction band minimum (CBM) are also shown.

**Table 1 | Two-component density functional theory calculated positron state lifetimes (ps)**

| bulk | $V_{Sb(1)}^{-3}$ | $V_{Sb(2)}^{-3}$ | $V_{Se(1)}^{-2}$ | $V_{Se(2)}^{-2}$ | $V_{Se(3)}^{-2}$ | $V_{Se(1)}^{0}$ | $V_{Se(3)}^{0}$ | $(V_{Sb(1)}V_{Se(2)})^{-1}$ | $(V_{Sb(1)}V_{Se(2)})^{-3}$ | $(V_{Sb(2)}V_{Se(1)})_{short}^{-1}$ | $(V_{Sb(2)}V_{Se(1)})_{short}^{-3}$ |
|------|------|------|------|------|------|------|------|------|------|------|------|
| 257 | 330 | 314 | 311 | 311 | 302 | 329 | 332 | 343 | 338 | 328 | 325 |

Here we report the detection of vacancy–related defects in thin film and in the near surface region of single crystal $Sb_2Se_3$ using VE-PALS measurements performed with the high intensity positron beamline (NEPOMUC) at the Heinz Maier–Leibnitz Zentrum (MLZ) research reactor in Garching[29,30]. Calculations of positron lifetimes using projector augmented-wave (PAW) TC-DFT for perfect lattice $Sb_2Se_3$ and relevant vacancy defects are also reported. In addition, binding and formation energy calculations for possible divacancy defects are performed. The presence of monovacancy defects with characteristic positron state lifetimes consistent with Se vacancies and Sb vacancies is observed in n-type samples. An absence of positron trapping to monovacancy defects is observed in the p-type sample. Divacancy defects were also observed in some samples. The results are consistent with DFT study reported by Wang et. al.[6], and performed in this work.

## Results

### TC-DFT of the characteristic positron lifetime values

The positron lifetime values were calculated for annihilation from the perfect lattice state in $Sb_2Se_3$, and for states localized at neutral and negatively charged monovacancies and for possible divacancy defects. They were made using ABINIT with the PAW method[28,31]. Calculations were performed for the −3 charge state of the two possible Sb monovacancy defects and the predicted neutral and −2 charge states for the possible Se vacancies. These calculations started from the ground-state defect configurations obtained by Wang et al.[6].

In the case of the divacancies, we performed a full search of stable geometries by combining different first-neighbour Sb and Se mono-vacancies in varying charge states. The initial structures were generated by removing pairs of Sb and Se atoms from the inequivalent sites, then the neighbour atoms were displaced using the SHAKENBREAK algorithm[22]. The structure was then relaxed within the DFT calculation. The resulting divacancy binding and formation energies are shown in

Fig. 2 (also see Supplementary Note 1). The −1 and −3 charge states of the two divacancies $V_{Sb(1)}V_{Se(2)}$ and $V_{Sb(2)}V_{Se(1)}$ in the short-bond configuration exhibit the lowest binding and formation energies. The resulting calculated positron state lifetime values for monovacancy and relevant divacancy defects in $Sb_2Se_3$ are given in Table 1 (Also see Supplementary Table 1).

A perfect lattice (bulk) positron lifetime, $\tau_b$, of 257 ps was obtained. The resulting positron densities for the perfect lattice state and for the positrons state localized at $V_{Sb(1)}^{-3}$ are shown in Fig. 3. The calculated monovacancy localized positron state lifetimes are well separated from the bulk lifetime (Table 1), they were found to lie in the range 302 ps to 332 ps while the values for the most stable divacancies spanned the range 325 ps to 343 ps (Also see Supplementary Note 1).

Significant differences in the lifetimes for vacancy defects at the different possible inequivalent sites were observed. The lifetime for the −3 charge state of the Sb monovacancy was found to be 314 ps for site-2 but increased to 330 ps for the more open configuration at site-1. The resolution of positron lifetime spectroscopy is limited, hence, if Sb vacancies are present at both sites it would be expected that the experimental lifetime spectrum would exhibit a single lifetime component due to trapping at these defects that would be the weighted average of the contributions from the two sites and hence would have a lifetime value intermediate between 314 ps and 330 ps.

Lifetime values were also calculated for positron states localized at the proposed neutral and −2 charge states for the Se vacancies at site-1 and site-3 and for the −2 charge state for the site-2 Se vacancy starting from the geometries reported by Wang et al.[6]. The resulting Se vacancy −2 state lifetimes span from 302 ps for site-1 to 311 ps for site-2 (Table 1). The lifetime values for the neutral charge states of the site-1 and site-3 Se vacancy were notably longer at 329 ps and 332 ps, respectively.

These calculations provide evidence that positron lifetimes in the approximate range 300−310 ps indicate trapping to −2 state Se

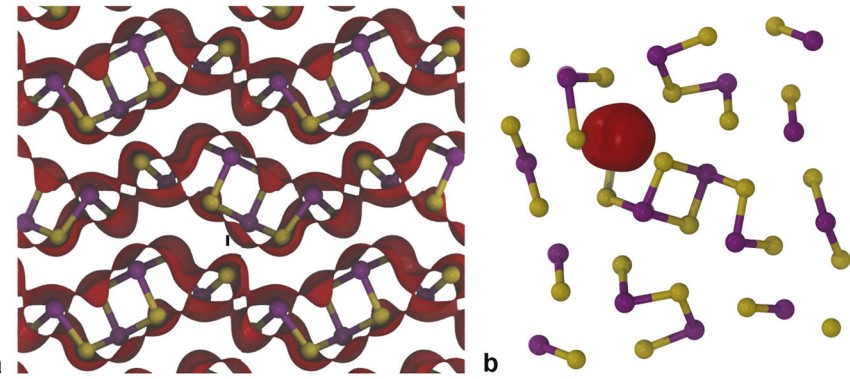

**Fig. 3 | Positron density in perfect Sb₂Se₃ and at Sb vacancy.** Two-component density functional theory calculated positron density isosurface (plotted at 20 % of the maximum value) shown in red, **a** for the delocalised perfect lattice state, and **b** localized at the Sb(1) vacancy in the −3 charge state.

monovacancies, lifetimes in the approximate range 315 to 330 ps are consistent with trapping to −3 charge state Sb monovacancies. However, lifetimes around 330 ps are also consistent with the less probable neutral state of the Se monovacancy. The positron states of the most stable divacancy defect ($V_{Sb(1)}V_{Se(2)}$) have a lifetimes of 343 ps and 338 ps for the −1 and −3 charge states, respectively.

**Variable energy positron lifetime spectroscopy measurements**
These were performed on a series of single crystals prepared by Bridgman melt growth and included samples with p-type, intrinsic, and n-type conductivities[32,33] (Supplementary Table 2). Measurements were also made on n-type Sb₂Se₃ thin films deposited using closed-space sublimation (CSS) and thermal evaporation (TE). Positron lifetime spectra were measured using different positron implantation energies. Typically, 4 keV, either 6 or 8 keV, and either 14, 16 or 17 keV energies were used and example positron implantation profiles are shown in Fig. 4a. For the 4 keV implantation energy almost all positrons are stopped in the top 150 nm and the mean implantation depth is ~63 nm, for 8 keV all are implanted to depths less than 450 nm and the mean depth is ~209 nm, while for 16 keV the mean depth is ~511 nm and almost all are implanted depths down to 1100 nm (Fig. 4a).

The experimental positron lifetime spectra were deconvolved assuming a finite number of discrete lifetime states. Fig. 5 shows example spectra with fits for a p-type, Sn-doped, Sb₂Se₃ crystal and an n-type CSS thin film. Almost all the spectra measured fitted to three lifetimes, see Supplementary Tables 3 and 4 and Fig. 4, with the first lifetime component having a value less than or approximately equal to the TC-DFT calculated perfect lattice lifetime of 257 ps while the second component lifetime value was normally in the range 301(7) ps to 354(1) ps, consistent with trapping to monovacancy and or divacancy defects. The third component typically had a lifetime in the range 0.6–6 ns with an intensity less than 4% and is due to a small fraction of positrons forming positronium (Ps) (Supplementary Note 3). The standard trapping model predicts that when positrons are trapping to a population of vacancy defects, the first lifetime component should have a value less than the perfect lattice lifetime: this is termed the reduced bulk lifetime component, and its lifetime value and intensity systematically decrease with increasing trapping to vacancy defects (Supplementary Note 3)[25,26]. Observation of a spectrum component with a lifetime value greater than the perfect lattice lifetime, but less than the minimum Ps lifetime (~550 ps), unambiguously demonstrates positron trapping to neutral or negatively charged vacancy-related defects.

The lifetime components obtained by fitting the spectra from a p-type Sb₂Se₃:Sn crystal, and from two Sb₂Se₃ crystals exhibiting intrinsic conductivity, are given in Supplementary Table 3 and are shown in Fig. 4b. The spectra measured from the p-type crystal were dominated by a first lifetime component comparable to, or shorter than, the TC-DFT calculated positron lifetime of 257 ps for the perfect lattice state (Table 1). The 8 and 14 keV spectra yielded a first lifetime of 260 ps and 261 ps, respectively, with an intensity of ~95% consistent with almost all positrons annihilated from the perfect lattice state. However, it was only for the near-surface spectrum, where positrons are implanted to a mean depth of 63 nm (4 keV, Fig. 4a), that a vacancy-related defect component was detected. This had a lifetime value of 343(8) ps ($I_2 = 38(5)$ %) (Fig. 4b, Supplementary Table 3) but the spectrum was dominated by the first component at 239(4) ps ($I_1 = 62(5)$%), the reduced bulk lifetime component. Assuming the values in Table 1 are accurate, the observation of a component with a lifetime greater than ~330 ps requires the presence of vacancy defects larger than monovacancies. Further, the defect lifetime of 343(8) ps is in agreement with the TC-DFT calculated value for the most stable divacancy defect configuration ($V_{Sb(1)}V_{Se(2)})^{-1}$ providing evidence for positron trapping to vacancy-related defects dominated by divacancies.

For the p-type crystal, positron trapping to vacancy-related defects was only observed in the near-surface (4 keV) spectrum, where the mean implantation depth was ~63 nm. The spectra measured using 8 keV and 14 keV were dominated by perfect lattice annihilations (Fig. 4b) indicating an absence of trapping to monovacancy or divacancy defects. The result is consistent with either vacancy-related defects restricted to only a narrow near surface region and being in neutral or negative charge states, or with a more uniform, deeper, distribution of defects but with those below the top layer existing only in positive charge states. Variable energy x-ray photoemission spectroscopy measurements have been performed on a similar p-type Sn-doped crystals and demonstrated the presence of n-type inversion layer below ~13 nm[34]. The existence of this layer would be consistent with the observation of positron trapping to divacancy defects only for the 4 keV spectrum, assuming these defects are neutral or negative for Fermi level positions close to or above midgap but positively charged for Fermi level positions in the lower half of the bandgap. First principles calculations predict the monovacancy defects in Sb₂Se₃ exhibit this behaviour[6], and Fig. 2a predicts the divacancy defects are not stable for Fermi level positions towards the VBM.

The positron lifetime results from the two nominally stoichiometric Sb₂Se₃ crystals that exhibited intrinsic conductivity (Supplementary Table 2) are also shown in Fig. 4b (Supplementary Table 3). The 8 keV spectrum from crystal A, where all positrons implanted to depths less than or equal to ~450 nm (Fig. 4a), was dominated by trapping to a vacancy-related component with a lifetime of 316(4) ps ($I_2 = 71(3)$ %). Comparison with the TC-DFT calculated values (Table 1) shows it to be significantly smaller than the divacancy lifetimes and to be comparable to the value for $V_{Sb(2)}^{-3}$, providing evidence for the

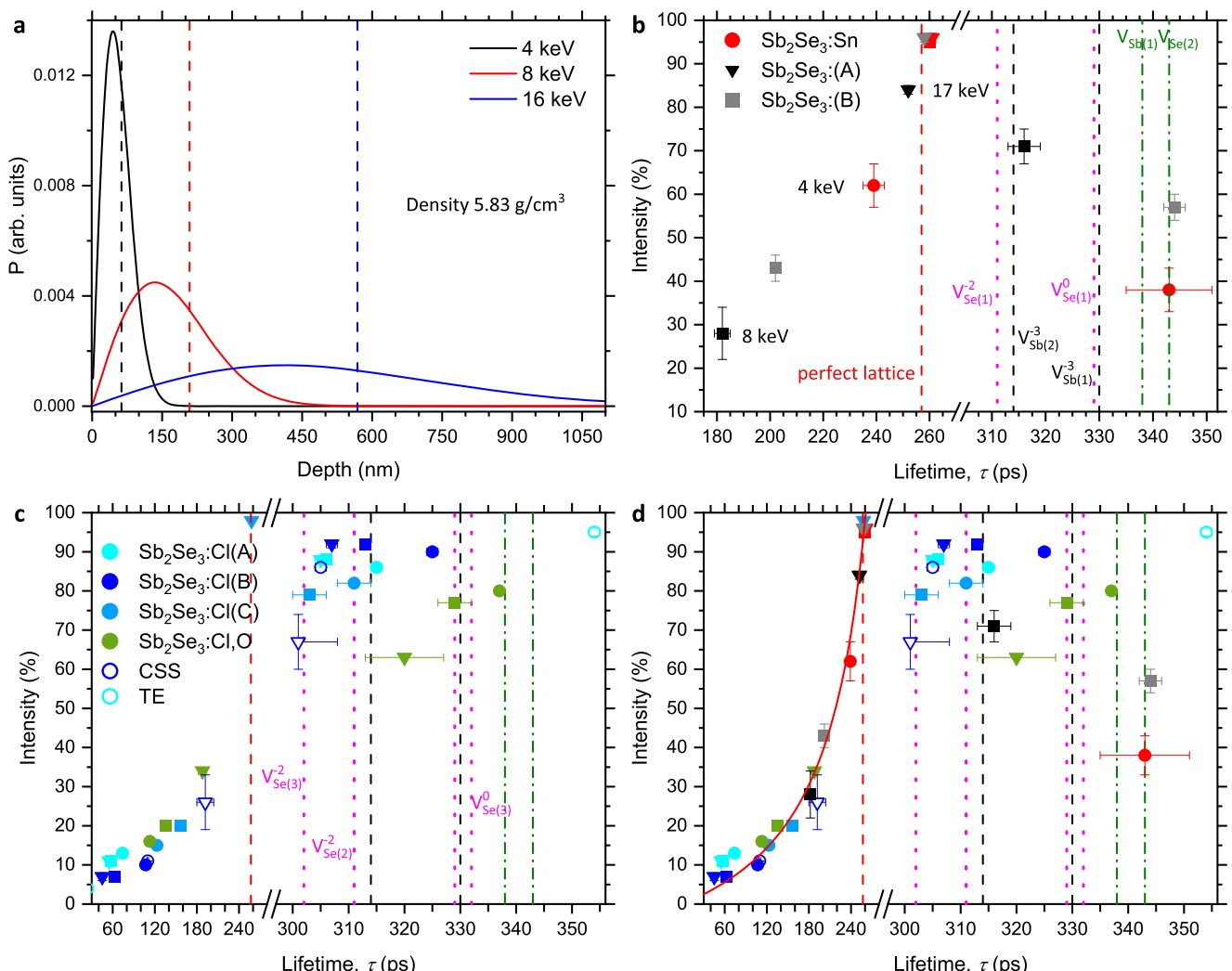

**Fig. 4 | Variable energy positron lifetime measurements. a** Markovian positron implantation depth profiles, the mean implantation depths are shown with dashed lines. **b** The first two deconvolved positron lifetime components for the p-type Sb₂Se₃:Sn and the two intrinsic Sb₂Se₃ crystals. Spectrum components measured using 4 keV implantation energy are shown as circles, for 6 or 8 keV by squares and for 14, 16 or 17 keV measurements by down triangles. The calculated lifetimes are shown using dashed lines. **c** The spectrum components obtained from the n-type Sb₂Se₃:Cl and Sb₂Se₃:Cl,O crystals and the closed-space sublimation and thermal evaporation deposited thin films. **d** Spectrum components from all samples. A standard trapping model fit to the reduced bulk component intensity assuming trapping to a single defect with a lifetime of 325 ps is shown as a guide (solid red line). The fit standard deviation error bars are shown.

presence of monovacancy defects. However, the 14 keV spectrum from crystal A was dominated by annihilations from the perfect lattice state but with minority component with a lifetime 467(11) ps ($I_2 = 15(1)$ %) longer than that predicted for divacancy defects, providing evidence for the presence of a low concentration of larger multivacancy cluster defects within the top micron of the sample. The 6 keV spectrum from intrinsic conductivity crystal B also showed positron trapping to vacancy-related defects, while the 17 keV spectrum was again dominated by annihilations from the perfect lattice state. For crystal B the lifetime of the defect component observed in the sub-450 nm region was 344(3) ps ($I_2 = 57(2)$ %), longer than that observed for crystal A, and is consistent with positron trapping to the most stable divacancy defect configuration ($(V_{Sb(1)}V_{Se(2)})^{-1}$ (Table 1). Nevertheless, the higher implantation energy spectra from both intrinsic conductivity crystals, which probe the top ~1200 nm, yielded spectra dominated by annihilations from the perfect lattice state (Fig. 4b and Supplementary Table 3).

The positron lifetime results for the n-type crystal and thin film samples are shown in Fig. 4c and Supplementary Table 4. Three Sb₂Se₃ crystals grown from commercially supplied Sb₂Se₃ found to contain Cl

contamination and to exhibit n-type conductivity were studied. Crystals Sb₂Se₃:Cl(A) and Sb₂Se₃:Cl(B) had been subjected to post-growth anneal in a Sb poor ambient. The 4 keV spectra from (A) and (B) returned a dominant lifetime component with values of 315(1) ps ($I_2 = 89(1)$ %) and 325(1) ps ($I_2 = 89(1)$ %), respectively, consistent with the presence of Sb monovacancies within the top 150 nm of the samples. However, the spectra measured at 8 keV and 14 keV gave dominant defect components with values of 306(1) ps ($I_2 = 88(1)$ %) and 305(1) ps ($I_2 = 88(1)$ %) for (A), while (B) returned 313(1) ps ($I_2 = 92(1)$ %) and 306(1) ps ($I_2 = 92(1)$ %), respectively. Comparing these experimental lifetime values to the TC-DFT calculated values in Table 1 provides evidence for dominant positron trapping to −2 charge state Se monovacancies in the region below the near surface layer.

The third crystal Sb₂Se₃:Cl(C) came from a different growth batch and was subjected to a post-growth anneal in a Se-poor environment (Supplementary Table 2). The 4 keV and 8 keV spectra were dominated by components with lifetimes of 311(3) ps ($I_2 = 82(1)$ %) and 303(3) ps ($I_2 = 79(2)$ %) again indicating dominant trapping to −2 charge state Se monovacancies. But the high energy (17 keV) spectrum was dominated by perfect lattice annihilation (258(3) ps, $I_1 = 98(1)$ %), providing

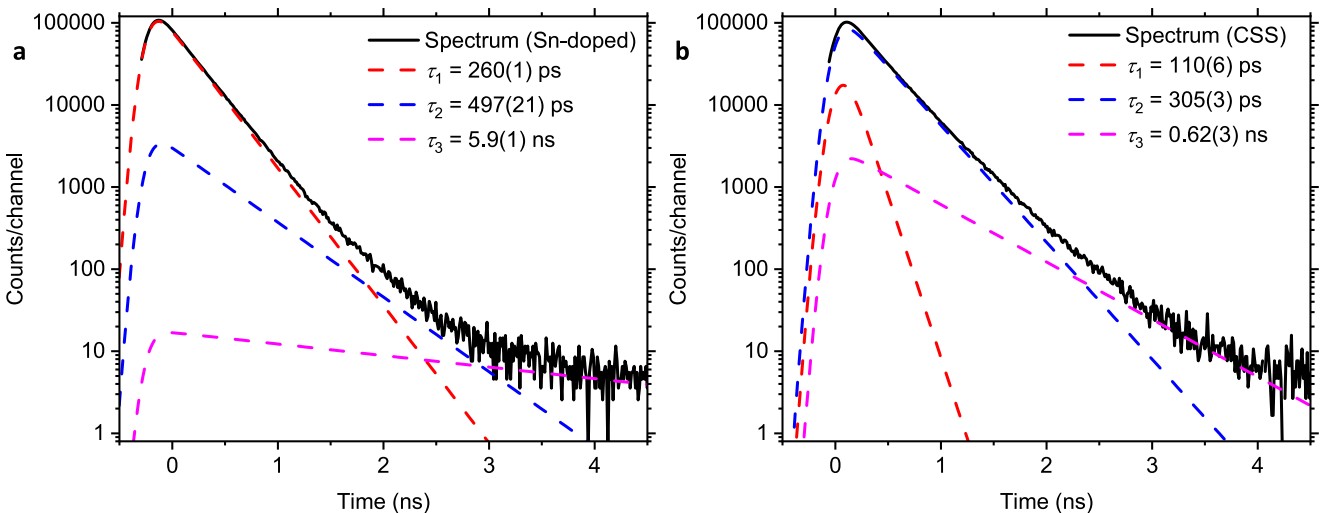

**Fig. 5 | Experimental positron lifetime spectra.** Also showing the deconvolution fit components. **a** The 8 keV positron implantation energy spectrum for a p-type, Sn-doped, Sb₂Se₃ crystal. **b** Closed-space sublimation deposited n-type conductivity Sb₂Se₃ thin film measured with a 4 keV positron implantation energy.

evidence that the monovacancies were present only in the top ~450 nm of the sample.

An oxygen doped crystal, Sb₂Se₃:Cl,O, exhibiting n-type conductivity was also studied (Fig. 4c, Supplementary Table 4). The 4 keV near surface spectrum was dominated by a defect component with value 337(3) ps ($I_2 = 81(1)$ %) indicating the presence of divacancy defects. The 6 keV and 17 keV spectra, however, were dominated by components with values of 329(3) ps ($I_2 = 77(1)$ %) and 320(7) ps ($I_2 = 73(4)$ %), respectively. These values are consistent with positron trapping to −3 charge state Sb monovacancies, but the 329(3) ps value is also in agreement with calculated values for the site-1 and site-3 neutral charge state Se monovacancies and overlaps the divacancy defect range. However, it can also be noted that the observed n-type conductivity suggests the Fermi level should be above the 0/−2 charge transition levels for the Se monovacancies[6].

A closed-space sublimation grown Sb₂Se₃ thin film and a thermally evaporated thin film were also studied. Both samples exhibited n-type conductivity due to Cl incorporation. The 4 keV and 16 keV spectra from the CSS thin film (Fig. 4c and Supplementary Table 4) were both dominated by a defect component with a lifetime of ~305 ps, consistent with trapping to −2 charge state Se monovacancies. The intensity of the component reduced from 86(1) % to 67(7) % with increasing implantation energy. The thermal evaporated film was only measured at 4 keV, the spectrum was dominated by a defect component with a lifetime of 354(1) ps. This is close to, but longer than, divacancy defect values, suggesting that trapping is dominated by divacancies but that there are also contributions from trapping to larger open-volume defects.

The dominant lifetime components from all the spectra measured are shown in Fig. 4d. There is a clear, systematic variation in the lifetime and intensity of the first component. The lifetime and intensity increase toward limiting values as the lifetime approaches ~260 ps. There is a concomitant decrease in the intensity of the second, vacancy-related defect, lifetime component. This behaviour is expected from the standard trapping model (STM) (Supplementary Note 3). The values of the bulk lifetime calculated from the experimental spectra using the STM are given in Supplementary Tables 3 and 4, and the average of these gave 260(6) ps. There is good agreement between this experimental value for the bulk, perfect lattice, state lifetime and the TC-DFT calculated value of 257 ps. There is also satisfactory agreement between the range of possible monovacancy lifetimes

calculated by TC-DFT (Table 1) which span from 302 ps for $V_{Se(3)}^{-2}$ to 330 ps for $V_{Sb(1)}^{-3}$ and the lifetime values for the second vacancy-defect related component obtained from the large majority of the spectra measured which range between 301(7) ps and 329(3) ps (Fig. 4d). However, it can be noted that if the less stable $V_{Sb(2)}V_{Se(1)}$ divacancies (Fig. 2a) were present the calculated −1 and −3 state lifetime values of 328 ps and 325 ps, respectively (Table 1), would overlap with part of this range. Four spectra returned longer second lifetime values consistent with the TC-DFT calculated lifetime range for divacancy defects. These were measured using either 4 keV or 6 keV implantation energies for which all positrons were implanted to depths less than 150 nm or 300 nm, respectively.

Vacancy-related defect lifetime components with lifetime values in the range 301 ps to 313 ps were only observed from some of the n-type samples (Fig. 4c). The TC-DFT calculations attribute lifetimes in this range to positron states localized at −2 charge state Se monovacancies (Table 1). This is consistent with the recently proposed unusual amphoteric behaviour for the Se monovacancies at the three possible sites, all of which are predicted to have a charge transition level to the −2 charge state in the upper half of the bandgap[6]. Lifetime components with values in the range 315–329 ps were observed from n-type samples and an intrinsic conductivity crystal (Fig. 4). This is the lifetime range associated with the −3 charge state Sb monovacancies. For those spectra that return a component value consistent with the neutral Se vacancies it is not possible to differentiate between the different assignments based on lifetime measurements alone. However, it should be noted that the neutral charge states of the Se vacancies are predicted to be stable for only a narrow range of Fermi level positions just above midgap, while the −3 charge for Sb monovacancies is predicted for a wide range of level positions above midgap[6]

The detection of a reduced bulk lifetime component in measured spectra enables the rate of positron trapping to vacancy defects, $\kappa_D$, to be estimated (Supplementary Note 3). The vacancy defect concentration is related to the trapping rate, $[V] = \kappa_D/\mu_V$, where $\mu_V$ is the defect specific trapping coefficient. The values of $\mu_V$ for negatively charged vacancy defects in various semiconductors have been reported to be in the range[25] ~0.1 × 10¹⁵–30 × 10¹⁵ s⁻¹. Using the typical value of ~3 × 10¹⁵ s⁻¹ the resulting estimated vacancy defect concentration obtained from the 8 keV spectrum from crystal Sb₂Se₃:(A) is ~1 × 10¹⁶ cm⁻³ while the 4 keV spectrum from crystal Sb₂Se₃:Cl(B) gives a value of ~4 × 10¹⁶ cm⁻³ (Supplementary Note 3).

## Discussion

In summary, we studied p-type, intrinsic, and n-type $Sb_2Se_3$ samples using depth-dependent positron annihilation lifetime spectroscopy. The deconvolved spectra normally comprise two dominant lifetime components, and the behaviour of the lifetime values and intensities is in good agreement with the standard positron trapping model. The results determine an experimental value for the perfect lattice positron state lifetime of 260(6) ps. Two-component density functional theory calculated a value of 257 ps (Table 1), in good agreement with experiment. The majority of the measured spectra yielded a second lifetime component with a lifetime in the range 301–354 ps unambiguously demonstrating positron trapping to vacancy-related defects in those samples. A significant minority of the spectra were dominated by annihilations from the perfect lattice, bulk, positron state demonstrating an absence of trapping to neutral or negative vacancy-related defects in the region sampled by the implanted positrons.

The TC-DFT calculations obtained the lifetimes for the positron states localized at the two Sb vacancy sites and the three Se vacancies for the possible charge states, and associated local geometries, reported by Wang et al. [6]. These yielded values ranging from 302 ps to 329 ps (Table 1). In addition, calculations were performed for the most stable divacancy defects, which gave lifetimes from 325 ps to 343 ps. The range of possible monovacancy and divacancy trapped positron state lifetime values obtained by TC-DFT is in good agreement with the range of experimental vacancy-related defect component lifetime values; 301–354 ps. This agreement, along with the agreement between the experimental and calculated perfect lattice state lifetime, provides evidence supporting the accuracy of the TC-DFT calculated lifetimes.

Lifetime components in the range 337(3) ps to 354(1) ps (Fig. 3d) were observed from four of the samples and provide clear evidence for the presence of divacancy defects (Table 1). These were observed in either 4 keV or 6 keV spectra and so were restricted to the near-surface regions of those samples.

The majority of the measurements on n-type $Sb_2Se_3$ samples (Fig. 4c, Supplementary Table 4) returned lifetime values in the range 301(7) ps to 313(1) ps consistent with the TC-DFT calculated values (Table 1) for positrons trapped at Se monovacancies in the −2 charge state. Positron lifetime components with values in the range 315(1) ps to 325(1) ps were measured from n-type samples and the intrinsic crystal A (316(4) ps) (Fig. 3, Supplementary Tables 3 and 4). These values are consistent with the TC-DFT calculated lifetimes for the two −3 charge state Sb monovacancies.

The results presented support the predictions from the DFT calculations that employed the SHAKENBREAK method to determine the ground state configurations of the possible monovacancy defects in $Sb_2Se_3$[6]. In particular, the existence of a stable −2 charge state for the Se vacancies for Fermi level positions in the upper portion of the bandgap. The predicted local geometries were found to give TC-DFT calculated positron state lifetimes that were shorter than those for the −3 charge state for the Sb vacancies, and experimental lifetimes in this range were observed in n-type samples. Further, the prediction that both Se and Sb monovacancies are in positive charge states for Fermi level positions below midgap is consistent with the experimental observation of the absence of positron trapping in the p-type crystal below the near-surface inversion layer.

Our experimental study, supported by first principles calculations, shows that positron annihilation lifetime spectroscopy can detect, identify, and quantify both Se and Sb monovacancy defects when in negative charge states and that larger divacancy defects can also be present in $Sb_2Se_3$. Depth-profiling positron lifetime measurements can enable direct detection of these point defects in $Sb_2Se_3$ solar cell device materials.

## Methods

### Sample preparation

The $Sb_2Se_3$ single crystal and thin film samples are detailed in Supplementary Note 2 and summarized in Supplementary Table 2. The crystal samples were grown by the Bridgman method in evacuated quartz capsules, which had been back-filled with argon to a pressure of 400 Torr to suppress macroscopic gas bubble formation. Experiments were conducted on their (100) cleavage surfaces and for which the ribbons lie on the b-axis, i.e., parallel to the cleavage plane (using the *Pnma* setting of Space Group 62)[35]. All elemental and compound source materials were from Alfa Aesar. Crystals $Sb_2Se_3$ (A) and (B) were grown from source material that was formed in-house by direct combination of the elements (6 N Sb and 5 N Se) heated quartz capsules. The chlorine doped crystal ($Sb_2Se_3$:Cl) was commercially supplied Alfa Aesar 5 N purity $Sb_2Se_3$, which has been shown to have unintentional chlorine doping at the level of ~$4 \times 10^{16}$ cm$^{-3}$ rendering it n-type[36]. The oxygen doped sample ($Sb_2Se_3$:Cl,O) was formed by adding 5 N $Sb_2Se_3$ to 5 N Alfa Aesar $Sb_2Se_3$ at the level of 0.01% by formula units ($Sb_2O_3/Sb_2Se_3$). Since the starting material also contained chlorine (see above), this sample may be considered to be co-doped with both chlorine and oxygen. It was also subjected to an extended annealing treatment to make it Sb-rich in order to encourage n-type conduction (isothermal heating with 55 at% Sb/45% Se at 340 °C for 336 hrs). Source material for a tin, doped selenium-rich crystal ($Sb_2Se_3$:Sn) was prepared by direct combination of the elements (6 N Sb and 5 N Se and Sn), with the tin being 0.01% of the antimony.

The $Sb_2Se_3$ thin films were formed by Close Space Sublimation (CSS) onto $SnO_2$:F coated soda lime glass (Pilkington TEC15), the source material being Alfa-Aesar 5 N $Sb_2Se_3$, i.e., chlorine doped, as above. The films were approximately 1500 nm in thickness. The thermally evaporated (TE) films were fabricated using similar source material placed in crucible in a chamber evacuated to $10^{-4}$ mbar and heated to high temperature to vapourise the source onto the same type of substrates. The TE films were approximately 800 nm in thickness.

### Positron annihilation

Variable energy positron annihilation lifetime spectroscopy was performed using the PLEPS instrument on the NEPOMUC high intensity beam line at the Heinz Maier–Leibnitz Zentrum (MLZ) research reactor in Garching[29,30]. Spectra contained $4 \times 10^6$ counts. The spectra were fitted using the software package PALSfit Version 3.195 (Technical University of Denmark, Riso Campus)[37]. The timing instrument resolution function was determined using a SiC standard sample, which was described by three Gaussian functions; the resulting full-width half-maximum values varied from 210 ps to 280 ps for the measurements described.

### Theory calculations

To support the experimental observations, two types of first-principles calculations were performed. To explore the stability of divacancies, we calculated their formation and binding energies using the SHAKENBREAK method[22,38] and the hybrid PBE0 functional as implemented in CP2K[39,40]. These calculations were used to identify the lowest-energy geometries and charge states of the Sb−Se divacancies. Further computational details are provided in Supplementary Note 1. To allow for the interpretation of the positron annihilation lifetime measurements, we carried out two-component density functional theory (TC-DFT) calculations using the projector-augmented-wave method in ABINIT[28,31]. The electron−positron correlation was treated using the Boronski−Nieminen functional with the Barbiellini et al. gradient correction[41,42] within the GGA-PBE framework. Lifetimes were obtained for the perfect lattice, the relevant Sb and Se monovacancies, and the

lowest-energy divacancy configurations. Additional methodology information is given in Supplementary Note 1.

## Data availability

The positron lifetime data that support the findings of this study and source data for display items have been deposited in figshare with the identifier 10.6084/m9.figshare.c.8192384. The processed data used in this study is provided within the paper and the Supplementary Information file.

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

## Acknowledgements

D.J.K. gratefully acknowledges the financial support provided by FRM-II to perform the high-intensity positron beam measurements at Heinz Maier-Leibnitz Zentrum (MLZ), Garching, Germany. J.W. acknowledges funding from the "Area of Advance - Materials Science" at Chalmers University of Technology and the Swedish Research Council (2019-03993). The computations were partly performed on resources provided by the Swedish National Infrastructure for Computing (SNIC) at NSC and PDC. J.M. and K.D. would like to acknowledge support from EPSRC grants EP/N014057/1m and EP/M024768/1. M.D. and W.E. gratefully acknowledges BMBF-grants 05K13WN1-POSIANALYSE, 05K16WN1-POSITEC and 05K19WN1-POSILIFE of the German Federal Office of Research and Education.

## Author contributions

D.J.K. with K.D., J.M., and T.D.C.H. designed the study. K.D. and T.D.C.H. supplied samples. D.J.K., M.D. and W.E. performed the positron annihilation experiments. D.J.K. and M.R.M.E. fitted the experimental spectra. J.W. and E.B. performed the DFT and TC-PAW-DFT calculations. D.J.K., with help from J.W., T.D.C.H. and K.D., wrote the main draft. All authors commented on the manuscript.

## Competing interests

The authors declare no competing interests.
