## [Transparent Peer Review file · Nature Communications]

Detection and identification of vacancy defects in antimony selenide

Corresponding Author: Professor David Keeble

Version 0:

Reviewer comments:

Reviewer #1

(Remarks to the Author)

In the submission, the authors report that vacancy-related defects in Sb₂Se₃ was studied by depth-dependent positron annihilation lifetime spectroscopy. They also calculated the positron lifetime of the associated defects. As a result, they provide the evidence for the observation of Sb monovacancies in the -3 state and of Se monovacancies in the -2 state. At first glance, the manuscript appears to be well written, but after careful reading, it was observed that some points require more detailed discussion and examination. The novelty required for publication in "Nature Communications" is also limited in the paper. As a result, it is recommend a rejection of this paper.

1. The positron annihilation technique was used to detect the vacancy defects of Sb₂Se₃, and this gives us a better understanding of vacancy defects in Sb₂Se₃ materials. But the authors seem to have only tested for defects in Sb₂Se₃, lacking further scientific explanation of the different nature of defects in different samples, much less guidance on regulating defects in Sb₂Se₃.

2. The authors do not provide any other results on the properties of the sample other than PALS tests, including specific test results on the type of conductivity, so we doubt whether the tested sample has the N-type, intrinsic, or P-type conductivity described by the authors. Authors should provide more information about the samples to ensure the authenticity and reliability of the study.

3. It is also a question whether there really are Se vacancies in -2 states. Many theoretical calculations of the defects of Sb₂Se₃, including its sister material Sb₂S₃, indicate that the anionic vacancy defect is either positively charged or neutral, and only the reference 6 in your submission mentions that V_{Se} is an amphoteric defect. How does the author explain this?

4. The drawings in the manuscript need to be drawn more carefully. Both Picture 1 and Picture 2 seem to be a screenshot taken directly from the software. Because the figure1 has a little horizontal line under the direction marker.

5. In Line 42 of Page 2, the authors said "Antimony selenide solar cell PCEs have increased from, for example, 2.3% in 2014 to 7.6% in 2018, then from 9.2% in 2019 to 10.6% in 2022-5", but none of the solar cells in the articles you cited have an efficiency of 10.6%; The authors cited the wrong references.

6. Why is the positron lifetime of neutral Sb vacancies not calculated in table 1?

7. In line 407 of page 21, the authors mentioned that "Lifetime components in the range 337(3) ps to 354(1) ps (Fig. 3d) were observed from four of the samples and provide clear evidence for the presence of divacancy defects (Table 1)." Why there is no monovacancy defects but divacancy defects in the P-type sample, which goes against our common sense; There needs to be a sound scientific explanation.

8. In line 327 of page 17 and line 338 of page 18, the name of the referenced figure is written incorrectly. Figure 3b should be Figure 3c.

9. In line 291 of page 16, "The positron lifetime results from the two nominally stoichiometric Sb₂Sb₃ crystals that exhibited intrinsic conductivity (Supplementary Table 3) are also shown in Fig. 3b (Supplementary 292 Table 4)." The authors misspelled Sb₂Se₃ instead of Sb₂Sb₃.

Reviewer #2

(Remarks to the Author)

In this manuscript, the authors investigate and identify open volume defects with neutral and negative charges in different Sb₂Se₃ samples by the combination of positron annihilation lifetime spectroscopy and density functional theory. They claim

the presence of monovacancy and divacancy in intrinsic and n-type samples, and the absence of monovacancy defects in p-type Sb₂Se₃. In my opinion, additional analysis and clarifications are needed to support the conclusions and make the work suitable for publication.

Below are specific comments for the authors to consider in revising the manuscript.

1. The authors focus on vacancies, but the concentrations of antisites with open volumes may also be substantial in this material. Could they also contribute to the observed signals? It should be clarified in the text if the observed signal is due to a concentration-weighted average over all possible defect species and an assumption is made to only consider vacancies.
2. The authors distinguished the monovacancy and divacancy by comparing the experimental positron lifetimes with the DFT results. However, the separation between calculated lifetimes between monovacancy and divacancy is vague. For unambiguous defect identification, additional techniques, such as Doppler broadening spectroscopy which offers insights into the local chemical environment of defects, could be beneficial.
3. The positron lifetime is sensitive to the concentration of each defect species. Please discuss how robust the results are for changes in defect concentration by plotting a figure similar to Fig. 6a in Physical Review B, 2019, 99(6): 064108. I am concerned that the defect assignment may be influenced by this choice.
4. The configurations and formation energies of divacancies appear to be absent from the report and should be included.

Minor

5. On the 32th line of page 2 the authors stated "The results are consistent with recent density function theory predictions that both the Sb and the Se monovacancy defects are amphoteric". However, as the present study only considers neutral and negatively charged defects, so the conclusion of "amphoteric" cannot be drawn.

6. Other points:

- On the 126th line of page 6, "These methods have unique specificity for vacancy, open volume, defects and have a sensitivity limit of approximately 10¹⁵ cm⁻³":
 - o These methods -> This method
 - o Please specify what the limit of 10¹⁵ cm⁻³ is for
 - o A citation should be added for the limit here
- On the 160th line of page 7, combining -> combining
- On the 9th line of SI, please correct the repetitive sentences
- On the 35th line of SI, please cite for "the Standard Trapping Model (STM)"

Reviewer #3

(Remarks to the Author)

The manuscript "Detection and identification of vacancy defects in antimony selenide" by David J. Keeble and co-authors reports a major step forward in the identification of vacancy defects in antimony selenide, that is a highly promising rapidly emerging material for future generation solar cells, with reported lab record efficiencies above 10% achieved in the last few years. Keeble and co-authors convincingly demonstrate that the presence of Sb and Se monovacancies and divacancies can be uniquely resolved in Sb₂Se₃ by combining state-of-the-art positron annihilation lifetime spectroscopy studies and ab-initio calculations (including local relaxation of atoms due to the densities of the electrons and positron). These studies were performed at the expert level and support the conclusions well.

This paves the way to systematically examine the presence and concentrations of these point defects in Sb₂Se₃ layers tailored for application in solar cells, which is key to improve the current relatively modest open circuit voltages (around 50% of the Shockley-Queisser limit), related to the presence of point defects and extended defects leading to unwanted charge carrier recombination. The well-described research as well as the highly accessible explanation of the key concepts of positron annihilation spectroscopy and ab-initio calculations will make this work highly valuable generally for researchers working in the fields of semiconductor layers for solar cells or other applications where point defects play an important role in their functional properties.

I recommend publication of this highly interesting manuscript in Nature Communications, after some minor revisions, as suggested in my (technical) comments listed below, have been considered by the authors.

Questions and Comments:

1. Figure 2b, p. 10 (discussed on p. 8, line 165-167) shows the calculated positron density in the Sb(1) vacancy. Figure 1 on p. 4 suggests that the neighbouring Se(2) atom loses its single atomic bond. What happens with the Se(2) atom, does it go to an interstitial site or does it bind to another atom?
2. Table 1, p. 9: The calculated lifetime associated with a neutral Se(2) monovacancy is missing. Is there a special reason why this was left out from the study?

3. p. 11, lines 204-210: the ranges for discriminating between Se(2-) monovacancies, Sb(3-) monovacancies and divacancies are reported to within an accuracy of 1 ps. This is too accurate, since systematic experimental inaccuracies in the positron lifetime are around 3-5 ps [26]. Also systematic inaccuracies will exist for the calculated positron lifetimes of these point defects obtained from the ab-initio study. I would like to suggest to report more global ranges for discrimination of these defects: Se(2-) monovacancies ~300 to 310 ps, Sb(3-) monovacancies: ~315 to 330 ps, Se(0) monovacancies: around 330 ps, divacancies: ~330 to 350 ps. This leaves the conclusions well intact.

4. Figure 3d, p. 13, and text on p. 18, lines 344-346: The single trapping model is applied in a highly convincing manner, not only evident by the extracted bulk (perfect lattice) positron lifetime that systematically falls in the range of 260(6) ps, but also demonstrated by the dependence of I_{RB} as a function of the reduced bulk lifetime τ_{RB} . I would like to suggest to plot, as a guide-to-the-eye, the function $I_{RB}(\tau_{RB}) = (\tau_D/\tau_B - 1)/(\tau_D - \tau_{RB}) * 100\%$ of the single trap model, with $\tau_B=257$ ps and $\tau_D = 320$ ps (average value for the defects observed), that fits the measured the intensity I_{RB} very well for the relevant range in τ_{RB} of 100 to 257 ps.

Remarks/suggestions on technical details:

1. p. 2, line 44: $V_{oc} \sim 40\%$ of bandgap value is stated. It would be better to state that $V_{oc} \sim 50\%$ of Shockley-Queisser limit (since bandgap values cannot be reached theoretically, instead of the normally applicable Shockley-Queisser detailed balance limit).

2. p. 6, line 127: a sensitivity limit of 10^{15} cm^{-3} is quoted, that in fact depends on the material and type of vacancy (10^{16} cm^{-3} for the neutral vacancy and 10^{15} cm^{-3} for the negatively charge vacancy in c-Si). It seems appropriate to state here that such defects can be evaluated in the range of $\sim 10^{15} \text{ cm}^{-3}$ to $\sim 10^{19} \text{ cm}^{-3}$ [26].

3. p. 11, line 215-216: thermal evaporation (TE) is stated as one of the synthesis methods of the Sb_2Se_3 films, but this method is not specified in the Methods Section, p. 23, line 455. Also, the thicknesses of the films are not specified in the manuscript, while these are crucial for the interpretation of the positron lifetime results, with implantation energies ranging from 4 to 17 keV (up to 1000 nm depth).

4. Fig. 3b, p. 13: the label 17 keV for $\text{Sb}_2\text{Se}_3(\text{B})$ should be 6 keV.

5. p. 20, lines 380-383: vacancy concentrations in units of $\text{cm}^{-3}\text{s}^{-1}$ are specified, while in the Supplementary Information specific trapping coefficients μ_V in units of s^{-1} are provided. Were the unit cell dimensions of Sb_2Se_3 applied to convert the dimensions? A typical value of $\sim 6 \times 10^{-8} \text{ cm}^{-3}\text{s}^{-1}$ is used in the manuscript, while a typical value for μ_V of $6 \times 10^{15} \text{ s}^{-1}$ is stated in the Supplementary Information. The two values are not positioned in the same way within the ranges specified (for example a typical value of $\sim 1.2 \times 10^{-7} \text{ cm}^{-3}\text{s}^{-1}$ is expected, based on the typical value for μ_V).

6. Supplementary Information, p. 5, line 43: In equation (1), the – sign should be a + sign in the sum of the two terms.

Reviewer #4

(Remarks to the Author)

Version 1:

Reviewer comments:

Reviewer #1

(Remarks to the Author)

This study applies the positron annihilation technology to the selenide antimony material system to detect the presence of vacancy defects. In the revision, the authors have properly addressed previous concerns. Regarding the analysis, I have minor suggestions to this study.

It might be necessary to provide some additional information to further support the conclusion. For instance, in Figure 4, the author has presented two spectra of positron annihilation lifetimes. Could you provide the original curves of the positron annihilation lifetimes in other cases and the corresponding results of the three-exponential fitting?

Is the result of the three-index fitting of the lifetime of positron completely reliable? The authors mentioned that a longer positron lifetime (For example, 467 ns) indicates the presence of larger open volume defects, can it be considered that these are some holes present in the crystal? If this is the case, what is puzzling is that there are holes in the single-crystal samples, while the deposited film samples do not have any holes at all. This is highly contrary to common sense.

Apart from these issues, this study is indeed an interesting and novel one, and it holds significant importance for the future microscopic non-destructive detection of materials.

Reviewer #2

(Remarks to the Author)

While I appreciate that the authors have addressed Q1 and Q3 in sufficient technical detail, the evidence provided for divacancy identification (Q2 and Q4) remains insufficient. My concerns are as follows:

1. Formation energy not discussed:

The manuscript does not include any estimate of the formation energy of the divacancy, nor does it reference prior work on its thermodynamic stability in Sb_2Se_3 . Without this, it's difficult to evaluate whether such a defect is likely to form under the relevant growth or annealing conditions. Even if the calculated positron lifetime matches the experimental value, a divacancy with a high formation energy would likely exist only in negligible concentrations.

2. Lifetime difference within uncertainty margins:

Given the close proximity of the calculated lifetimes, it is challenging to reliably distinguish the divacancy from the monovacancy based on positron lifetime alone. The lowest predicted divacancy lifetime is nearly indistinguishable from the highest monovacancy value, while the highest divacancy lifetime exceeds it by only about 20 ps. This difference falls well within the typical combined uncertainty of positron lifetime measurements and TC-DFT calculations, making it difficult to unambiguously attribute the longer lifetime component to a divacancy.

3. Limited exploration of charge states and configurations:

The divacancy geometries were generated by directly combining two monovacancies. However, it is not clear whether other plausible divacancy configurations or charge states were considered. This limits the strength of the comparison between theory and experiment, as both the structure and charge state can significantly affect the formation energy and positron lifetime. I would expect at least a brief discussion of this aspect.

Overall, these points suggest that the divacancy identification should be discussed with more caution, as the current wording suggests a level of certainty that the evidence does not fully support.

Reviewer #3

(Remarks to the Author)

The authors of the manuscript "Detection and identification of vacancy defects in antimony selenide" satisfactorily answered my questions and comments, explaining well their views and insights on these points. They carefully and properly revised the manuscript including the Supplementary Information. They addressed the comments of all reviewers. I recommend publication of this highly interesting manuscript in its current version in Nature Communications.

Minor text suggestions:

1. p. 7, line 144: "Here we report the detection of vacancy-related defects" (add 'of')

2. p. 22, line 420: "In particular, this concerns the prediction of a stable" (add 'this concerns' to complete the sentence)

3. Supplementary Information, p. 10, line 4: "in vacuum.6" (It concerns Reference 6 of the list of References on the same page)

Reviewer #4

(Remarks to the Author)

Version 2:

Reviewer comments:

Reviewer #1

(Remarks to the Author)

In this revision, the authors have well addressed my concerns, the manuscript is recommended for acceptance.

Reviewer #2

(Remarks to the Author)

I am satisfied with the response. The additions to the manuscript have made it a more substantial piece of work, and I am happy to recommend publication without further changes.

Reviewer #4

(Remarks to the Author)

I co-reviewed this manuscript with one of the reviewers who provided the listed reports. This is part of the Nature

Communications initiative to facilitate training in peer review and to provide appropriate recognition for Early Career Researchers who co-review manuscripts.

We thank the reviewer for their careful consideration of the manuscript and for their constructive comments and corrections. They have enabled us to produce a higher quality, more accurate, revised manuscript. We address each comment in the below response.

Response to the remarks of Reviewer #1

The positron annihilation technique was used to detect the vacancy defects of Sb_2Se_3 , and this gives us a better understanding of vacancy defects in Sb_2Se_3 materials. But the authors seem to have only tested for defects in Sb_2Se_3 , lacking further scientific explanation of the different nature of defects in different samples, much less guidance on regulating defects in Sb_2Se_3 .

The reviewer is referred to Supplementary Table 3 and the 'Methods: Sample preparation' section for details of the methods of used for controlling and regulating the defects in Sb_2Se_3 crystals. Here we explained that two methods were used for stoichiometric control: a) by weighing at the synthesis stage, with the stoichiometric ratios being given explicitly in column 3 of the table and b) by post-growth annealing under controlled vapour pressures of the elements as detailed in column 4.

Whilst method a) is self-explanatory, method b) relies on the manipulation of the vapour-solid equilibrium position. Experimentally this amounts to controlling both the temperature of the material itself and the vapour pressure of the vapour species above it so as to encourage either anion or cation vacancies. (The vapour-solid equilibrium relation is reviewed in Piacente, V.; Scardala, P.; Ferro, D. Total vapour-pressure of solid Sb_2Se_3 . *Journal of Materials Science Letters* **1992**, *11* (12), 855-857. DOI: 10.1007/bf00730486.) Hence the materials tested and their stoichiometric condition were selected to give a broad range of point defect outcomes: This is consistent with the aim of the study was to experimentally determine the range, identity and charge states of vacancy defects in Sb_2Se_3 for the first time.

The authors do not provide any other results on the properties of the sample other than PALS tests, including specific test results on the type of conductivity, so we doubt whether the tested sample has the N-type, intrinsic, or P-type conductivity described by the authors. Authors should provide more information about the samples to ensure the authenticity and reliability of the study.

The referee requests details of the measurement of conductivity type. We refer the referee to ref 36. The conductivity type was determined using the hot probe method with details of the method used described in ref 36. Where carrier concentrations are given in the text, they were measured using the Hall method, which also confirmed the carrier type. To further clarify the conductivity of the samples studied we have added a detailed description into Supplementary Note 2 and included the results from hot-probe measurements on the samples.

It is also a question whether there really are Se vacancies in -2 states. Many theoretical calculations of the defects of Sb_2Se_3 , including its sister material Sb_2S_3 , indicate that the anionic vacancy defect is either positively charged or neutral, and only the reference 6 in your submission mentions that VSe is an amphoteric defect. How does the author explain this?

Ref 6 (Wang XW, et al. *Phys Rev B* **108**, 134102 (2023)) from the Walsh group is the first high quality DFT study to implement the more comprehensive search algorithm seek a global local structure

minimum (also see Wang et al Joule 8 2105 (2024). We discuss this explicitly in the paragraph starting “The removal of an Sb atom....” on p5 and concluding on p6.

The drawings in the manuscript need to be drawn more carefully. Both Picture 1 and Picture 2 seem to be a screenshot taken directly from the software. Because the figure1 has a little horizontal line under the direction marker.

The incorporated images in Fig.1 and in Fig. 2 were jpegs but the image files supplied for publication are full resolution eps files.

In Line 42 of Page 2, the authors said “Antimony selenide solar cell PCEs have increased from, for example, 2.3% in 2014 to 7.6% in 2018, then from 9.2% in 2019 to 10.6 % in 2022-5”, but none of the solar cells in the articles you cited have an efficiency of 10.6%; The authors cited the wrong references.

We thank the reviewer for pointing out this typo, it should indeed have been 10.1% not 10.6%, this has been corrected.

Why is the positron lifetime of neutral Sb vacancies not calculated in table 1?

Because the available high quality density functional theory studies report that the Sb vacancies are not stable in the neutral charge state, they are negative-U centres, see e.g. Wang XW, et al. Phys Rev B 108, 134102 (2023). We discuss this explicitly by us in the paragraph starting “The removal of an Sb atom....” on p5 and concluding on p6.

In line 407 of page 21, the authors mentioned that “Lifetime components in the range 337(3) ps to 354(1) ps (Fig. 3d) were observed from four of the samples and provide clear evidence for the presence of divacancy defects (Table 1).” Why there is no monovacancy defects but divacancy defects in the P-type sample, which goes against our common sense; There needs to be a sound scientific explanation.

This is an important point. We make it very clear that we are not saying that there are no monovacancy defects in the p-type material but what we are saying is that if monovacancies are present they must be in positive charge states and hence cannot trap positrons. As we discuss in the manuscript, the study of Wang et al predicts that the Se and Sb vacancies will be in positive charge states in p-type material. We return to this point in our discussion.

A lifetime consistent with the presence of divacancies is only observed in the 4 keV spectrum (see Supplementary Table 4) where positrons are implanting to depths below 150 nm. In the manuscript we give an explanation for this. Experiments have shown the presence of narrow inversion layer at the surface of the p-type sample hence vacancy-related defects in this region can exist in neutral or negative charge states and trap positrons.

In summary, we are not saying there are no monovacancies in the p-type sample and we do provide sound scientific explanations for why we are observing a minority divacancy lifetime component in the 4 keV spectrum from the p-type sample.

In line 327 of page 17 and line 338 of page 18, the name of the referenced figure is written incorrectly. Figure 3b should be Figure 3c.

We thank the reviewer, and these typos have been corrected.

In line 291 of page 16, “The positron lifetime results from the two nominally stoichiometric Sb_2Sb_3 crystals that exhibited intrinsic conductivity (Supplementary Table 3) are also shown in Fig. 3b The authors misspelled Sb_2Se_3 instead of Sb_2Sb_3 .

We thank the reviewer, and this typo has been corrected.

Response to the remarks of Reviewer #2

In this manuscript, the authors investigate and identify open volume defects with neutral and negative charges in different Sb₂Se₃ samples by the combination of positron annihilation lifetime spectroscopy and density functional theory. They claim the presence of monovacancy and divacancy in intrinsic and n-type samples, and the absence of monovacancy defects in p-type Sb₂Se₃. In my opinion, additional analysis and clarifications are needed to support the conclusions and make the work suitable for publication.

Below are specific comments for the authors to consider in revising the manuscript.

1. The authors focus on vacancies, but the concentrations of antisites with open volumes may also be substantial in this material. Could they also contribute to the observed signals? It should be clarified in the text if the observed signal is due to a concentration-weighted average over all possible defect species and an assumption is made to only consider vacancies.

The positron lifetime results presented are unambiguously due to open volume vacancy defects. The method is essentially blind to antisite or interstitial defects.

While it is indeed true that an antisite defect in a negative charge state would result in a slight increase in positron density in the vicinity it does not strongly localise as it does at the site of a missing atom. The positron density attracted to a negative antisite defect, or similarly any negatively charged substitutional acceptor, essentially samples an environment that is similar to the perfect lattice and yields a lifetime component with a value at or close to the bulk lifetime (see K. Saarinen, et al. Phys. Rev. B **39**, 5287 (1989)). The presence of weak localisation at negative non-vacancy centres has been inferred from variable temperature measurements at low temperature due the departures from the normal standard trapping model behaviour due to these centres starting to complete more effectively with trapping to vacancies. However, even in this case it does not alter the lifetimes of any resolved vacancy defect trapping these will still have their characteristic values.

In the measurements presented here we are clearly observing vacancy trapping lifetimes, further the standard trapping model accurately describes the results (e.g. Fig 3d) which provides evidence that if there are antisite defects present they are not strongly influencing positron annihilation.

2. The authors distinguished the monovacancy and divacancy by comparing the experimental positron lifetimes with the DFT results. However, the separation between calculated lifetimes between monovacancy and divacancy is vague. For unambiguous defect identification, additional techniques, such as Doppler broadening spectroscopy which offers insights into the local chemical environment of defects, could be beneficial.

We agree with the reviewer, they highlight one of the significant outcomes from this manuscript. We demonstrate the viability of future coincidence Doppler broadening spectroscopy (CDBS) measurements. The observation of the perfect lattice positron lifetime in p-type samples will enable future studies to obtain precision CDBS ratio spectra. Further TC-DFT calculations are also required to enable this future study.

3. The positron lifetime is sensitive to the concentration of each defect species. Please discuss how robust the results are for changes in defect concentration by plotting a figure similar to Fig. 6a in Physical Review B, 2019, 99(6): 064108. I am concerned that the defect assignment may be influenced by this choice.

The statement “The positron lifetime is sensitive to the concentration of each defect species” is not in general correct, it requires a specification of which positron lifetime is being discussed. It is only true if the lifetime being referred to is the average positron lifetime, $\bar{\tau}$ (or a reduced bulk lifetime). The average positron lifetime is critically dependent on concentration of each defect species present since $\bar{\tau} = \sum_i I_i \tau_i$ and the defect concentrations determine the intensities. The positron state lifetimes for specific vacancy-related defects are independent of defect concentration and are characteristic of the local environment of the defect.

Unzueta et al (PRB 99 064108) were only able to report average lifetimes for their rather complex Heusler alloy samples. It seems they were unable to deconvolve their experimental spectra. These had modest statistics ($3E6$) and required subtraction of source correction components. In consequence, the only way they are able to attempt to infer defect concentrations was to assume they could use the standard trapping model (STM) expression $[V] = (1/\tau_B \mu_V) [(\bar{\tau} - \tau_B)/(\tau_V - \bar{\tau})]$ but they were also forced to use their TC-DFT calculated values for both τ_B and the value defect lifetime τ_V (and to assume there was only one positron trapping defect). Their Fig. 6a is just the textbook plot of the STM showing the variation of $\bar{\tau}$ with vacancy defect concentration for the one-defect case. It was only possible for them to infer possible defect concentration ranges by mapping their only experimental $\bar{\tau}$ values to the predictions from the above formula using their theoretical τ_B and τ_V values as seen in their Fig. 6b and Fig. 6c.

We are in a much more fortunate position. Our spectra have higher statistics and more importantly do not require source correction. In consequence we can deconvolve our lifetime spectra as shown in our Fig. 3 (and in the detailed tables in the Supplementary Information). We determine an *experimental* value for τ_B (Fig. 3d) and indeed happen to find the TC-DFT calculated value agrees with the experimental value. We directly determine an experimental vacancy defect lifetime from the deconvolution which provides direct evidence that we are dominantly in the one-defect case. The results show we are not in, or close to, the saturation trapping region and so can directly use the STM expression $[V] = \mu_V \kappa_V$ where for us κ_V is exclusively determined by the experimental values obtained from the deconvolution (see Supplementary Note 3 Eq. 3); this is in stark contrast to the situation of Unzueta et al.

The STM calculated concentration estimate we report is not sensitive to concentration in the same manner that mean lifetime is sensitive to concentration as reported in Unzueta et al (PRB 99 064108) Figs 6b and 6c. The defect component lifetime is intrinsically independent of concentration, of course its intensity and the values of the reduced bulk lifetime and its intensity are concentration dependent – this is what allows us to calculate the concentration.

4. The configurations and formation energies of divacancies appear to be absent from the report and should be included.

This manuscript describes the first experimental positron lifetime spectroscopy study of vacancy-related defects in Sb_2Se_3 and the necessary supporting two-component density functional theory (TC-DFT) calculations that enable interpretation. In contrast to ‘conventional’ DFT studies the primary focus of TC-DFT is the calculation of the positron parameters. These calculations use the output geometries from the Wang et al (2023) study (kindly shared by the authors) as initial inputs for the positron density and further structural relaxation optimisations. We are not reporting a full density function theory study of monovacancy and divacancy defects in this manuscript. Motivated by the clear

experimental observation of divacancies in certain samples reported here we do indeed plan a future combined theory and experimental study of divacancy and larger defects which would enable the calculation of formation energies and configurations, but this is outwith the scope of this study.

Minor

5. On the 32th line of page 2 the authors stated “The results are consistent with recent density function theory predictions that both the Sb and the Se monovacancy defects are amphoteric”. However, as the present study only considers neutral and negatively charged defects, so the conclusion of “amphoteric” cannot be drawn.

We agree with the reviewer and amphoteric has been omitted in the revised manuscript.

6. Other points:

- On the 126th line of page 6, “These methods have unique specificity for vacancy, open volume, defects and have a sensitivity limit of approximately 10^{15} cm^{-3} ”:
 - These methods -> This method
 - Please specify what the limit of 10^{15} cm^{-3} is for
 - A citation should be added for the limit here

The reason for the use of the plural is that the statement “These methods have unique specificity for vacancy, open volume, defects and....” is true for all positron annihilation spectroscopy methods positron lifetime, Doppler broadening spectroscopy and 2DACAR. The paragraph was aimed at a general audience; it starts with general statements before focusing on the method used here, positron lifetime spectroscopy.

The sensitivity limit is explicitly discussed in Reference 25 and is stated in Ref 26, we have now added these citations at the appropriate point.

The reviewer asks us to specify what type/s of open volume vacancy defect the approximate concentration sensitivity limit of 10^{15} cm^{-3} applies. This question has a rather complex and technical answer which due to the constraints of word count and of clarity for the non-specialist we did not articulate further. The standard positron trapping model (See Supplementary Note 3) states the trapping rate, κ_V , of positrons to an open volume defect is given by $\kappa_V = \mu_V[V]$ where $[V]$ is the defect concentration and μ_V is the defect-specific positron trapping coefficient. The sensitivity limit depends on the experimental limitations of measuring κ_V . In the simplest experimental situation of one positron trapping defect type then the sensitivity limit comes from the ability to detect and measure the intensity the defect lifetime component. The lower sensitivity limit for defect concentration is then given by product $[V]_{\min} = \kappa_{V\min}/\mu_V$, and hence also on the value of μ_V . The method has the best sensitivity for vacancy defects with the largest defect-specific positron trapping coefficients. Negatively charged vacancy defect will have a higher value than a neutral vacancy, the coefficient for a vacancy cluster defect can increase with increasing size, up to a certain point. The issue is our knowledge of the μ_V values is addressed in Supplementary Note 3.

- On the 160th line of page 7, ~~combing~~ -> combining

Correction made.

- On the 9th line of SI, please correct the repetitive sentences

The two sentences have been reworded.

- On the 35th line of SI, please cite for “the Standard Trapping Model (STM)”

Reference 5 is now cited.

Response to the remarks of Reviewer #3

The manuscript “Detection and identification of vacancy defects in antimony selenide” by David J. Keeble and co-authors reports a major step forward in the identification of vacancy defects in antimony selenide, that is a highly promising rapidly emerging material for future generation solar cells, with reported lab record efficiencies above 10% achieved in the last few years. Keeble and co-authors convincingly demonstrate that the presence of Sb and Se monovacancies and divacancies can be uniquely resolved in Sb₂Se₃ by combining state-of-the-art positron annihilation lifetime spectroscopy studies and ab-initio calculations (including local relaxation of atoms due to the densities of the electrons and positron). These studies were performed at the expert level and support the conclusions well.

This paves the way to systematically examine the presence and concentrations of these point defects in Sb₂Se₃ layers tailored for application in solar cells, which is key to improve the current relatively modest open circuit voltages (around 50% of the Shockley Queisser limit), related to the presence of point defects and extended defects leading to unwanted charge carrier recombination. The well-described research as well as the highly accessible explanation of the key concepts of positron annihilation spectroscopy and abinitio calculations will make this work highly valuable generally for researchers working in the fields of semiconductor layers for solar cells or other applications where point defects play an important role in their functional properties. I recommend publication of this highly interesting manuscript in Nature Communications, after some minor revisions, as suggested in my (technical) comments listed below, have been considered by the authors.

Questions and Comments:

1. Figure 2b, p. 10 (discussed on p. 8, line 165-167) shows the calculated positron density in the Sb(1) vacancy. Figure 1 on p. 4 suggests that the neighbouring Se(2) atom loses its single atomic bond. What happens with the Se(2) atom, does it go to an interstitial site or does it bind to another atom?

The primary purpose of our Figure 2b was to clearly demonstrate to a non-positron audience the positron localisation at vacancy-type defects. The precise details of the local structure of the site-3 – 3 charge Sb monovacancy is better illustrated in Wang et al (PRB 108 134102) FIG.2(a) (and more clearly in their supplementary information Figure I(b)). We thought it was not appropriate for us to go any further than we have into the level of local structural detail performed and described by Wang et al. There is a movement of the neighbour toward the vacancy site (Wang et al SI section S2).

2. Table 1, p. 9: The calculated lifetime associated with a neutral Se(2) monovacancy is missing. Is there a special reason why this was left out from the study?

Wang et al do not find a stable neutral charge state for the site-2 Se monovacancy, their calculations show it to be a strange four-electron negative-U centre flipping from +2 to – 2, in consequence we did not have a local geometry to enable a calculation.

3. p. 11, lines 204-210: the ranges for discriminating between Se(2-) monovacancies, Sb(3-) monovacancies and divacancies are reported to within an accuracy of 1 ps. This is too accurate, since systematic experimental inaccuracies in the positron lifetime are around 3-5 ps [26]. Also

systematic inaccuracies will exist for the calculated positron lifetimes of these point defects obtained from the ab-initio study. I would like to suggest to report more global ranges for discrimination of these defects: Se(2-) monovacancies ~300 to 310 ps, Sb(3-) monovacancies: ~315 to 330 ps, Se(0) monovacancies: around 330 ps, divacancies: ~330 to 350 ps. This leaves the conclusions well intact.

We completely agree with the reviewer and have restructured the paragraph and modified the lifetime ranges as suggested.

4. Figure 3d, p. 13, and text on p. 18, lines 344-346: The single trapping model is applied in a highly convincing manner, not only evident by the extracted bulk (perfect lattice) positron lifetime that systematically falls in the range of 260(6) ps, but also demonstrated by the dependence of I_{RB} as a function of the reduced bulk lifetime τ_{RB} . I would like to suggest to plot, as a guide-to-the-eye, the function $I_{RB}(\tau_{RB}) = (\tau_D/\tau_B - 1)/(\tau_D - \tau_{RB}) * 100\%$ of the single trap model, with $\tau_B = 257$ ps and $\tau_D = 320$ ps (average value for the defects observed), that fits the measured the intensity I_{RB} very well for the relevant range in τ_{RB} of 100 to 257 ps.

We thank the referee for this suggestion and have added a guide to the eye line to Fig. 3d and have added the appropriate expression with a brief explanation to Supplementary Note 3.

Remarks/suggestions on technical details:

1. p. 2, line 44: $V_{oc} \sim 40\%$ of bandgap value is stated. It would be better to state that $V_{oc} \sim 50\%$ of Shockley-Queisser limit (since bandgap values cannot be reached theoretically, instead of the normally applicable Shockley-Queisser detailed balance limit).

The referee is correct about the SQ limit - even though most papers refer to V_{oc} as a fraction of bandgap. For clarity we have therefore amended the manuscript to say it both ways.

Original manuscript states: "low values of open-circuit voltages, V_{oc} , which are typically only ~40 % of the bandgap value."

Changed to: "'low values of open-circuit voltages, V_{oc} , which are typically only ~40 % of the bandgap value (i.e. ~50% of the Shockley-Queisser limit)."

2. p. 6, line 127: a sensitivity limit of 10^{15} cm^{-3} is quoted, that in fact depends on the material and type of vacancy (10^{16} cm^{-3} for the neutral vacancy and 10^{15} cm^{-3} for the negatively charge vacancy in c-Si). It seems appropriate to state here that such defects can be evaluated in the range of $\sim 10^{15} \text{ cm}^{-3}$ to $\sim 10^{19} \text{ cm}^{-3}$ [26].

Indeed, we were trying to keep this issue of sensitivity as simple as possible for the non-specialist reader. We have considered the reviewers helpful suggestion and modified the sentence adding references.

Changed to: "These methods have unique specificity for vacancy, open volume, defects and have a sensitivity limit of approximately 10^{15} cm^{-3} and can estimate vacancy defects concentrations up to approximately 10^{19} cm^{-3} beyond which saturation positron trapping occurs^{25,26}."

3. p. 11, line 215-216: thermal evaporation (TE) is stated as one of the synthesis methods of the Sb₂Se₃ films, but this method is not specified in the Methods Section, p. 23, line 455. Also, the thicknesses of the films are not specified in the manuscript, while these are crucial for the interpretation of the positron lifetime results, with implantation energies ranging from 4 to 17 keV (up to 1000 nm depth).

We have corrected these omissions in the revised text. The CSS film was approximately 1500 nm in thickness and the thermally evaporated film was 800 nm. A description of the thermally evaporated films has been added to the Methods section.

4. Fig. 3b, p. 13: the label 17 keV for Sb₂Se₃(B) should be 6 keV.

The 17 keV label has been moved to the appropriate symbol and the symbol for the Sb₂Se₃ (B) data point corrected.

5. p. 20, lines 380-383: vacancy concentrations in units of cm⁻³s⁻¹ are specified, while in the Supplementary Information specific trapping coefficients μ_V in units of s⁻¹ are provided. Were the unit cell dimensions of Sb₂Se₃ applied to convert the dimensions? A typical value of $\sim 6 \times 10^{-8}$ cm⁻³s⁻¹ is used in the manuscript, while a typical value for μ_V of 6×10^{15} s⁻¹ is stated in the Supplementary Information. The two values are not positioned in the same way within the ranges specified (for example a typical value of $\sim 1.2 \times 10^{-7}$ cm⁻³s⁻¹ is expected, based on the typical value for μ_V).

We thank the reviewer for bringing this to our attention. We had made an error on p20 lines 380-383. In this sentence we should have quoted defect specific trapping coefficient values per atom per second, rather than the volume specific values in units of cm⁻³s⁻¹. For the semiconductor community there is a strong temptation to quote in units of cm⁻³s⁻¹, but as we are making a general point here we should have used s⁻¹.

We are quoting from Table 3.3 in Ref. 25 (Krause-Rehberg) and for negative charge state vacancies the range should strictly have been 0.1E15 to 30E15 s⁻¹. However, Krause-Rehberg & Leipner go on to quote a more selected range of experimental values along with theory values in Table 3.4. The average of these values for negative vacancies $\sim 3E15$ s⁻¹, this is a more probable value. We have modified the relevant text on p20 accordingly. We have also modified the text on p5 of Supplementary Note 3 to be consistent with the main text. We have used the 3E15 s⁻¹ value and calculated the resulting per volume trapping coefficient values for both the Sb and the Se sites in Sb₂Se₃ and obtain an approximate average value of 1.6×10^{-7} cm³s⁻¹. We have added text stating this in the modified Supplementary Note 3 and have changed the value of the two vacancy defect concentrations reported on p20 of the main text. This has resulted in a small reduction in these values compared to those given in the original manuscript.

6. Supplementary Information, p. 5, line 43: In equation (1), the – sign should be a + sign in the sum of the two terms.

This typo has been corrected.

We again thank the reviewers for their thorough and careful consideration of the manuscript and for their constructive comments and corrections. New Density Functional Theory calculations have been performed and are included to address directly the question raised regarding the binding and formation energies of the possible divacancies. The postdoctoral associate who performed these extensive additional calculations has, in consequence, been added to the author list.

We address each comment in the below response.

Response to the remarks of Reviewer #1

This study applies the positron annihilation technology to the selenide antimony material system to detect the presence of vacancy defects. In the revision, the authors have properly addressed previous concerns. Regarding the analysis, I have minor suggestions to this study.

It might be necessary to provide some additional information to further support the conclusion. For instance, in Figure 4, the author has presented two spectra of positron annihilation lifetimes. Could you provide the original curves of the positron annihilation lifetimes in other cases and the corresponding results of the three-exponential fitting? Is the result of the three-index fitting of the lifetime of positron completely reliable?

We chose to illustrate the results of the fitting of the experimental lifetime spectra along with the fit and fit components using two typical spectra in Figure 5 (Figure 4 previously), the plots for all the spectra look very similar to these and so we have not shown them all. We do include all the best fit results, including all the relevant three-component fits, in Supplementary Tables 3 and 4. We have updated these tables to include the reduced chi-squared values for each to show that the values are within the typical accepted reliability range.

The authors mentioned that a longer positron lifetime (For example, 467 ns) indicates the presence of larger open volume defects, can it be considered that these are some holes present in the crystal? If this is the case, what is puzzling is that there are holes in the single-crystal samples, while the deposited film samples do not have any holes at all. This is highly contrary to common sense.

The lifetime for a divacancy in this material is ~340 ps. The lifetime of 467 ps is consistent with trapping to a multivacancy cluster containing more than two vacancies. If the open volume was the size of a nanovoid the lifetime value would be in positronium range of $\sim 0.5 < \tau_{0-Ps} < 142$ ns (Supplementary Note 3). We have changed the text to clarify this point; “..a low concentration of larger multivacancy cluster defects..”.

Apart from these issues, this study is indeed an interesting and novel one, and it holds significant importance for the future microscopic non-destructive detection of materials.

We thank the reviewer for their comments which have contributed to our improved revised manuscript and for their positive assessment of that manuscript.

Response to the remarks of Reviewer #2

While I appreciate that the authors have addressed Q1 and Q3 in sufficient technical detail, the evidence provided for divacancy identification (Q2 and Q4) remains insufficient. My concerns are as follows:

1. Formation energy not discussed:

The manuscript does not include any estimate of the formation energy of the divacancy, nor does it reference prior work on its thermodynamic stability in Sb_2Se_3 . Without this, it's difficult to evaluate whether such a defect is likely to form under the relevant growth or annealing conditions. Even if the calculated positron lifetime matches the experimental value, a divacancy with a high formation energy would likely exist only in negligible concentrations.

We performed new extensive density functional theory calculations of the binding and formation energies of the possible divacancy configurations to address this point. The results are now included in the new Figure 2 and related new text in the manuscript (also see Supplementary Note 1). These show that VSb(1)VSe(2) divacancy has a binding energy of -0.81 eV and is predicted to have the lowest formation energy for Fermi level positions through most of the band gap. The subsequent TC-DFT calculations returned a positron state lifetime of 343 ps and 338 ps, for the -1 and -3 charge states (Table 1), respectively, in good agreement the experimental lifetimes (Fig. 4, Supplementary Tables 3 and 4). The VSb(2)VSe(1)-short configuration has a slightly lower binding energy while the other possible divacancy configurations have noticeably lower binding energies, more restricted ranges of stable Fermi level position, and higher formation energies (Fig. 2, Supplementary Note 1).

2. Lifetime difference within uncertainty margins: Given the close proximity of the calculated lifetimes, it is challenging to reliably distinguish the divacancy from the monovacancy based on positron lifetime alone. The lowest predicted divacancy lifetime is nearly indistinguishable from the highest monovacancy value, while the highest divacancy lifetime exceeds it by only about 20 ps. This difference falls well within the typical combined uncertainty of positron lifetime measurements and TC-DFT calculations, making it difficult to unambiguously attribute the longer lifetime component to a divacancy.

While there indeed systematic uncertainties in the TC-DFT calculations dependent on calculation method and approximations assumed the trend in lifetime values from perfect lattice to monovacancy to divacancy are normally reliable. In this work a centrally important result is the good agreement between the experimental perfect lattice state lifetime value of 260(6) ps which it unambiguously determined in this work and the TC-DFT calculated value of 257 ps. This provides strong supporting evidence that, possibly fortuitously, the method and approximations used are indeed yielding positron lifetime values appropriate for Sb_2Se_3 . This point is clearly made in the manuscript.

The experimental positron lifetime results unambiguously exhibit lifetime less than 320 ps (301 ps to 316 ps) from ten of the spectra (Fig. 4, Supplementary Tables 3 and 4). Our updated divacancy TC-DFT calculations performed using the ShakeNBreak/DFT optimised ground state geometries return divacancy positron state lifetimes ranging from 320 ps to 343 ps (Supplementary Table 1). The 320 ps value is for the least probably, most unstable, divacancy configuration (Fig. 2). A proposal that attributes the experimental lifetimes in the range ~300-320 ps to divacancy defects would not seem reasonable.

In performing the DFT calculations for the divacancies at the suggestion of the reviewer it can be noted that we also performed check ShakeNBreak/DFT calculations for the monovacancy defects, the results for the ground state configurations were in very good agreement with those of Wang XW et al (PRB 108 134102) (see Supplementary Fig. 1). The TC-DFT calculated monovacancy positron state

lifetimes range between 302 ps and 332 ps (Table 1). The longer values of 329 ps and 332 ps are for the lower probability neutral charge state of the Se vacancy (PRB 108 134102 Fig. 1b). Our experimental positron lifetime results return a positron lifetime between 337 ps and 354 ps for four of the spectra, two of which give values of ~ 343 ps (Fig. 4, Supplementary Tables 3 and 4). These lifetimes are unambiguously longer than 332 ps and a proposal that attributes these defect lifetimes to monovacancy defects would not seem reasonable.

As discussed in the manuscript experimental lifetime components in the range ~ 320 ps to 329 ps returned in three of the spectra cannot be unambiguously attributed to monovacancy defects due to the overlap with between the VSb(3) value of 330 ps with the range for the less probably neutral VSe states and the lower binding energy divacancy configurations (Table 1, Supplementary Table 1). Nevertheless, given that the -3 charge states of the Sb vacancies have the lowest formation energies (PRB 108 134102 Fig. 1b, Supplementary Fig. 1) for possible positron trapping vacancy-related defects it would not seem unreasonable to postulate, as we do, that these experimental lifetimes are more probably attributable to Sb monovacancies.

3. Limited exploration of charge states and configurations: The divacancy geometries were generated by directly combining two monovacancies. However, it is not clear whether other plausible divacancy configurations or charge states were considered. This limits the strength of the comparison between theory and experiment, as both the structure and charge state can significantly affect the formation energy and positron lifetime. I would expect at least a brief discussion of this aspect.

We now report full ShakeNBreak/DFT calculations that find the fully relaxed possible divacancy binding energies, formation energies, and local geometries, see answer to point 1.

Overall, these points suggest that the divacancy identification should be discussed with more caution, as the current wording suggests a level of certainty that the evidence does not fully support.

We appreciated the reviewers concerns on this point which is why we undertook to perform an exhaustive set of new DFT calculations to carefully explore the feasibility of the possible divacancy configurations and then use these ground state structures for the TC-DFT calculations. We are grateful to the reviewer for guiding us down this path as it has resulting in a far stronger, more rigorous, manuscript. It turned out the agreement between the lowest energy, most probably, divacancy configuration and our experimental results was also significantly improved.

Response to the remarks of Reviewer #3

The authors of the manuscript “Detection and identification of vacancy defects in antimony selenide” satisfactorily answered my questions and comments, explaining well their views and insights on these points. They carefully and properly revised the manuscript including the Supplementary Information. They addressed the comments of all reviewers. I recommend publication of this highly interesting manuscript in its current version in Nature Communications.

Minor text suggestions:

1. p. 7, line 144: “Here we report the detection of vacancy-related defects” (add ‘of’)

Correction made.

2. p. 22, line 420: “In particular, this concerns the prediction of a stable” (add ‘this concerns’ to complete the sentence)

We agree the English could be improved for this sentence, the subject – the DFT predictions – are referred to in the previous sentence and the repetition of prediction in the target sentence was also not ideal. In consequence we have modified the sentence to read “In particular, the existence of a stable –2 charge.....”.

3. Supplementary Information, p. 10, line 4: “in vacuum.6” (It concerns Reference 6 of the list of References on the same page).

Correction made.

We thank the review for their knowledgeable and highly positive assessment.